# Patterns of Brain Sparing in a Fetal Growth Restriction Cohort

**DOI:** 10.3390/jcm11154480

**Published:** 2022-08-01

**Authors:** Jon G. Steller, Diane Gumina, Camille Driver, Emma Peek, Henry L. Galan, Shane Reeves, John C. Hobbins

**Affiliations:** Division of Maternal Fetal Medicine, Department of Obstetrics and Gynecology, University of Colorado School of Medicine, Aurora, CO 80045, USA; diane.gumina@cuanschutz.edu (D.G.); camille.elizabeth.driver@gmail.com (C.D.); emma.peek@cuanschutz.edu (E.P.); henry.galan@cuanschutz.edu (H.L.G.); shane.reeves@cuanschutz.edu (S.R.); jchobbins@gmail.com (J.C.H.)

**Keywords:** fetal growth restriction, intrauterine growth restriction, middle cerebral artery, small for gestational age, uteroplacental insufficiency, vertebral artery

## Abstract

Objective: Our objective was to compare differences in Doppler blood flow in four fetal intracranial blood vessels in fetuses with late-onset fetal growth restriction (FGR) vs. those with small for gestational age (SGA). Methods: Fetuses with estimated fetal weight (EFW) <10th percentile were divided into SGA (*n* = 30) and FGR (*n* = 51) via Delphi criteria and had Doppler waveforms obtained from the middle cerebral artery (MCA), anterior cerebral artery (ACA), posterior cerebral artery (PCA), and vertebral artery (VA). A pulsatility index (PI) <5th centile was considered “abnormal”. Outcomes included birth metrics and neonatal intensive care unit (NICU) admission. Results: There were more abnormal cerebral vessel PIs in the FGR group versus the SGA group (36 vs. 4; *p* = 0.055). In FGR, ACA + MCA vessel abnormalities outnumbered PCA + VA abnormalities. All 8 fetuses with abnormal VA PIs had at least one other abnormal vessel. Fetuses with abnormal VA PIs had lower BW (1712 vs. 2500 g; *p* < 0.0001), delivered earlier (35.22 vs. 37.89 wks; *p* = 0.0052), and had more admissions to the NICU (71.43% vs. 24.44%; *p* = 0.023). Conclusions: There were more anterior vessels showing vasodilation than posterior vessels, but when the VA was abnormal, the fetuses were more severely affected clinically than those showing normal VA PIs.

## 1. Introduction

Fetal growth restriction (FGR) is associated with significant fetal morbidity and mortality. Early-onset FGR (diagnosed prior to 32 weeks) is characterized by severe oxygen and nutritional deprivation leading to elevated risks of neonatal morbidity and intrauterine fetal demise (IUFD). Late-onset FGR fetuses (diagnosed after 32 weeks of gestation) are still vulnerable to long-term neurological [1,2], cardiovascular [3,4], and metabolic morbidity [5]. In late-onset FGR, management considerations regarding the timing of Doppler studies, antepartum fetal heart rate testing, and delivery, vary appreciably. To differentiate fetuses with low estimated fetal weights (EFWs) who are at greatest risk for complications from those who are not overtly deprived, a group of international experts convened and drafted criteria via Delphi consensus [6] that have been endorsed by the International Society of Ultrasound in Obstetrics and Gynecology (ISUOG) [7]. ISUOG defines small for gestational age (SGA) as a fetus between the 3rd and 10th percentile with normal Dopplers and defines FGR as a fetus with an EFW or abdominal circumference (AC) < 3rd centile or an SGA fetus with abnormal Dopplers (as outlined in Table 1). Although the clinical value of the Delphi/ISUOG criteria for identifying fetuses at greatest risk of adverse outcomes has been well formulated [6], the concept has not been endorsed by some official bodies [8,9]. 

An important component of ISUOG criteria is the evaluation of the cerebroplacental ratio (CPR). This requires Doppler waveform assessments of the umbilical artery (UA), a reflector of placental resistance, and the middle cerebral artery (MCA), which reflects “brain sparing” as an adaptive reaction to counter fetal hypoxia [10]. Since the MCA perfuses the cerebral cortex, and its direction of flow within the brain lends itself to easy ultrasound sampling, it has been the principal source of information about the protective process of circulatory redistribution. However, other vessels emanate from the circle of Willis. The anterior cerebral artery (ACA), the posterior cerebral artery (PCA), and the vertebral artery (VA), which arrives in the brain via the subclavian artery on the right side and the aorta directly on the left, have been only sparsely investigated in the setting of FGR. There is evidence that these other vessels are involved in brain sparing and, in keeping with the concept of “gradual hypoxia” in FGR, may undergo vasodilatation at different times during the pathogenesis of mild-to-severe oxygen deprivation [11]. However, most investigations have been focused on the individual vessels [12,13,14] rather than how these vessels adapt collectively to “spare” the brain, especially in late-onset FGR, the more common form of clinical growth restriction.

Therefore, our objectives were: To determine how often each study vessel (MCA, ACA, PCA, and VA) had pulsatility index (PI) values below the 5th percentile in fetuses whose estimated fetal weights were below the 10th percentile and who were defined as having FGR or SGA using Delphi (ISUOG) criteria.To compare average cerebral vessel PI values in FGR and SGA fetuses with postnatal outcomes.To determine if any of the fetal cerebral vessels yielded additional useful information regarding perinatal outcome above that provided by the MCA as used in the CPR.

## 2. Methods

A prospective observational cohort study was undertaken in an off-campus university outpatient high risk center in Denver, CO, USA. Patients between 31 and 39 weeks gestational age were enrolled onsite from the center’s private referral base or from our on-campus perinatal practice. Inclusion criteria included fetuses with EFWs < 10th centile for gestational age with no obvious anatomical abnormalities and no absent/reversed end diastolic flow on umbilical artery Doppler assessment. Dating was established per ACOG (American College of Obstetricians & Gynecologists) criteria [15]. All initial dating was performed by the University of Colorado team unless the patients were referred from outside private clinics, for whom their dating was validated by our investigators at enrollment. The study was approved by the Colorado Multiple Institutional Review Board and informed consent was obtained from all study participants (IRB number 14-1360, date of approval 29 May 2015). 

Estimated fetal weights were calculated from measurements of the head circumference (HC), biparietal diameter (BPD), abdominal circumference (AC), and femur length (FL) using Hadlock nomograms [16]. Percentiles for EFW were assigned according to the population-based fetal growth curve by Hadlock [17], used consistently in our perinatal practice for its applicability to our study population. Following ISUOG guidelines, fetuses measuring less than the 10th percentile by EFW or AC were classified as either FGR or SGA (Table 1). Further, cerebral placental ratios (CPRs) were computed by dividing pulsatility indices (PI) of MCA and UA (MCA PI/UA PI), which were converted to percentiles using Ebbing’s nomogram [18,19,20]. Any fetus with an EFW or AC < 10th %ile with an abnormal umbilical artery Doppler (>95th %ile) or CPR (<5th %ile) or AC or EFW < 3rd percentile was defined as having FGR per ISUOG criteria. Those with EFWs between the 3rd and 10th percentiles with normal UA and MCA Dopplers were considered small for gestational age (SGA).

All examinations were performed during fetal quiescence. Both umbilical arteries were sampled from a free loop of umbilical cord, and the average PI was used for analysis. The cerebral vessels were imaged with directional color Doppler and waveforms obtained with pulsed Doppler. While an ideal angle of insonation for the UA and MCA of <20 degrees was obtained (Figure 1), a more liberal angle of <30 degrees was tolerated for the ACA, PCA, and vertebral arteries. All vessels emanating from the circle of Willis were sampled within the proximal third of the vessel, and although initially the near side vessel was preferentially chosen, the far side vessel occasionally provided a better waveform for analysis. The VA was approached posteriorly by obtaining a mid-sagittal view of the cervical spine and occiput (Figure 2). The vertebral artery most clearly visualized was sampled. Although only a small portion of the vessel can be imaged as it courses towards the posterior fossa just above the level of the first cervical vertebrae, the operator was aided by its distinctive directional Doppler signature (Figure 2). While every study parameter was attempted to be collected at each study visit, the results from the last, most complete exam before delivery were used for this analysis. Results from all 4 vessels were compared against control data from the literature, and the MCA, ACA, PCA, and VA were considered abnormal if the PI was <5th percentile using these control data [13,14,18,19,20,21]. All four vessels could not be perfectly visualized in all patients and some measures were excluded in images that were not of appropriate quality. Only the MCA was used in clinical decision-making such as the frequency of antepartum follow up and fetal heart rate testing, frequency of Doppler evaluation, or timing of delivery. 

A retrospective analysis was performed in this cohort of growth restricted pregnancies that had all clinical and Doppler information collected in a prospective fashion. Cerebral vessels from the Circle of Willis as well as birth outcomes were compared between FGR and SGA fetuses. Since the MCA, as used in the CPR, is the only cerebral vessel currently being used in FGR management, fetuses were further stratified according to whether the VA PI and CPR fell below the 5th centile. 

When evaluating outcomes, a cesarean delivery rate was calculated for all patients who attempted labor (i.e., not a scheduled repeat or elective cesarean section). We also reported rates of admission to the neonatal intensive care unit (NICU). 

Graphpad Prism Version 9 (San Diego, CA, USA) was used for statistical analyses. Fisher’s exact tests were used for categorical variables. Independent samples *t*-tests or Mann-Whitney U tests were used following a normality test to compare patient characteristics or Doppler measurements between groups. 

## 3. Results

During the study period, 81 patients were identified who met inclusion criteria. Thirty fetuses were classified as SGA and 51 patients as FGR. FGR pregnancies did not deliver significantly earlier than SGAs (37.3 weeks vs. 37.8 weeks, *p* = 0.186), but had significantly lower birthweights (2247 g vs. 2636 g, *p* < 0.0001) (Table 2). The average PIs of the ACA, MCA, and PCAs were all lower in FGR fetuses compared to SGA, but not enough to attain statistical significance (Table 2). The vessels most commonly having PIs below the 5th centiles in FGR were: the ACA (11/46), MCA (10/51), PCA (7/46), and VA (8/43) (Table 3). While the mean PI values were similar between the FGR and SGA groups, the number of abnormal PIs (<5th percentile) for the MCA and VA were significantly different between SGA and FGR groups (Table 3). In addition, whenever the VA PI was abnormal, it was never the only vessel showing brain sparing. Most of these fetuses (6/8) had abnormalities in 2 or more other cerebral vessels, and 3/8 had abnormalities in all 4 cerebral vessels.

In patients with an abnormal VA, the differences between the mean PI percentiles for all the other cerebral vessels were strikingly lower compared with those with normal VAs (Table 4). Further, newborns with an abnormal VA in utero delivered on average significantly earlier (35.2 versus 37.9 weeks, *p* = 0.0052), had lower birthweights (1712 g versus 2500 g, *p* < 0.0001), had a higher rate of cesarean section (CSR) (71.4% vs. 16.3% *p* = 0.005), and were more frequently admitted to the NICU (71.4% vs. 24.4% *p* = 0.023) (Table 4). Of fetuses admitted to the NICU within this study cohort, two were only for a brief transitionary period lasting less than 24 h; one of these fetuses with FGR with a normal CPR and normal VA PI, and the other fetus was SGA with a normal CPR and no VA PI measurement. All other admissions were greater than 24 h.

The mean PIs of the cerebral vessels were significantly lower in fetuses with an abnormal CPR compared to those with a normal CPR (Table 5). However, fetuses with an abnormal VA PI still had lower PIs in the other cerebral vessels than those fetuses with an abnormal CPR (Table 4 and Table 5). These findings strongly indicate that while the MCA is not alone in brain sparing activity, that the VA seemed to be associated with a greater degree of overall vasodilatation. For example, when the VA was abnormal, the PIs were lower for ACAs (3.4% versus 12.8%) and PCAs (7.7% versus 21.2%) compared with those fetuses having abnormal CPRs (Table 4 and Table 5). Furthermore, those with abnormal VAs delivered earlier on average (35.2 vs. 36.7 weeks) and had lower BWs (1712 g vs. 2009 g) than the fetuses with abnormal CPRs. However, when corrected for gestational age, the BW percentiles were similar (4.64% vs. 3.93%) (Table 4 and Table 5). Finally, those with abnormal VAs had a higher CSR and a higher percentage of admissions to the NICU (Table 4 and Table 5). As anterior cerebral vessel abnormalities were more frequently seen than in the posterior vessels (11 ACAs and 10 MCAs vs. 7 PCAs and 8 VAs) and fetuses with abnormal VA were the most severely affected with concomitant abnormalities in other vessels, the combination of these findings indirectly endorse the concept of front-to-back brain sparing in FGR (Table 3). 

## 4. Discussion

### Principle Findings

While fetuses defined by ISUOG as FGR were not associated with earlier deliveries than those defined as SGA (37.3 weeks vs. 37.8 weeks *p* = 0.186), they did have significantly lower average birthweights (2247 g vs. 2636 g, *p* < 0.0001) and lower birthweight percentiles (6.91% vs. 20.61% *p* < 0.0001). Although FGR fetuses were more frequently admitted to the NICU than SGA fetuses, this did not attain statistical significance.The total numbers of abnormal PIs from the cerebral vessels studied were strikingly different between SGA and FGR (4 vs. 36, *p* = 0.055). However, the number of cerebral vessel PIs below the 5th centile was only significant between FGR and SGA groups for the MCA and VA (Table 3). All 8 fetuses in the study with abnormal VA PIs were FGR.Fetuses with an abnormal VA PI stood out as being different in birth metrics and in their distinct tendency to be linked with lower PIs in each vessel emanating from the circle of Willis, compared with those with normal VA PIs (Table 4).When comparing in utero cerebral Dopplers and neonatal outcome data between fetuses with abnormal VAs and the commonly used CPR, the VA was associated with lower average PIs from the companion vessels, earlier delivery, lower birthweight, higher rates of cesarean section, and more frequent admission to the NICU, suggesting a more specific measure for adverse perinatal outcome than MCA. However, birthweight centiles were not different between groups when corrected for gestational age (Table 4 and Table 5).

In the early 2000s, investigators began to explore patterns of circulatory redistribution in the cerebral vessels in FGR [11,22,23,24] with one that utilized a sophisticated method to assess fractional moving blood volume [24]. These studies pointed to a front-to-back pathway of vasodilation with the frontal lobe receiving preferential initial attention, as evidenced by earlier or more frequent signs of ACA vasodilation than the MCA [11,23]. Our findings, using standard pulsed Doppler methods, support previous observations of a front-to-back pattern of brain sparing in FGR with those vessels feeding the anterior and middle portions of the brain frequently showing lower average PIs than those perfusing the posterior portion. When the VA PI was below the 5th centile, brain sparing had occurred in at least one other vessel for every fetus and in 3/8 patients all four cerebral vessels were abnormal. Assuming a front-to-back pattern of circulatory redistribution, by the time the VA had been compromised, all fetuses had lower average birthweights and birthweight centiles, earlier deliveries, and higher rates of admission to the NICU compared to other fetuses in the study (Table 4). Morales Roselló et al. [25] showed the vertebral artery to be more predictive of lower birthweight in FGR than the middle cerebral artery, a standard measurement in FGR management. The VA supplies the cerebellar hemispheres, vermis, and brain stem, and our findings suggest that fetuses with an abnormal VA may be at risk for more severe compromise and worse perinatal outcome. Furthermore, in these eight cases the average PIs (suggesting the degree of vasodilation and, therefore, hypoxia) were the lowest for all the three accompanying cerebral vessels studied, thus inferring a higher degree of hypoxia or deprivation. (Table 4).

FGR is a placentally-mediated condition. The effects on the fetus are dependent upon the degree of placental pathology and the ability of the fetus to adapt in the setting of placental insufficiency. Clinicians have historically depended upon the UA to indirectly assess the extent of placental pathology based on the concept that a substantial proportion of the small villus circulation of the placenta is compromised before the UA PI becomes abnormal [26]. In early-onset FGR, the placenta is so severely affected that an elevated UA PI is often one of the first clues to nutritional and hypoxic deprivation [27,28,29]. Yet, in the more common late-onset FGR, diagnosed after 32 weeks, the UA PI is frequently unaffected, even when cardiac deformation and abnormal myocardial contractility has been noted [30,31]. Thus, to help to distinguish between pathologic and ‘constitutional’ growth restriction, many clinicians have turned diagnostically to the MCA and, with it, the CPR (MCA PI/UA PI) to determine if undergrown fetuses have turned to brain sparing as a protective mechanism in dealing with fetal hypoxia. This approach has been effective in predicting the need for cesarean section for fetal distress [32], combined adverse neonatal outcome [33], and identifying those fetuses who are at greater risk for childhood and adulthood cardiovascular [3,4] and neurobehavioral morbidities [1,2]. Yet, the role of the MCA has not been uniformly accepted in the management of FGR [9], which has been primarily focused on avoiding intrauterine demise or severe perinatal morbidity via timely delivery predicated on information from the UA PI, fetal heart rate monitoring, and, in severe FGR, ductus venosus waveforms [34].

We addressed the question of whether any vessel or combination of vessels provided added value beyond that provided by the MCA or the commonly used CPR, in which the MCA contributes 50% to the result. Fetuses with abnormal VAs had lower average birthweights (1712 g vs. 2009 g), delivered at an earlier average gestational age (35.2 weeks versus 36.7 weeks), but had a slightly higher average birthweight centile (4.6% vs. 3.93%) compared to the average abnormal CPR BW percentile (Table 5). Despite the latter finding, there were 3 infants in the study who had abnormal VA PIs but whose CPRs were low but in normal range. Two of these infants appeared to be seriously compromised, having spent 5 and 26 days in the NICU and delivering at 36.14 and 33 weeks, respectively.

Although some information is available linking brain sparing, as indicated by MCA PIs linking to abnormal neurobehavioral outcomes, few studies have addressed the ability of other cerebral vessels to predict long-term outcomes [12,35]. One study did compare the vessel most often showing vasodilation in FGR (the ACA in their study) with the MCA in predicting neurobehavioral outcome in children using the Neonatal Neurobehavioral Adjustment Scale (NBAS). The authors found that more abnormalities were detected via abnormal MCA PI than ACA PI [12]. Our findings suggest that tracking long-term neurobehavioral abnormalities in fetuses with abnormal VA PIs might represent a more clinically important vessel for further investigation.

Another clinical finding revealed in Table 2 is that the SGA group had an average birthweight in the 20th centile as defined by the Fenton birthweight population curve used in our nursery and by many centers around the country. Since our entry criterion was simply to have EFWs below the 10th percentile, this result might question the accuracy of the in-utero ultrasound method to separate out undergrown fetuses. However, this type of neonatal curve cannot be cleanly used for comparisons to EFW because data from this neonatal population did not exclude fetuses affected by growth restriction, thereby pulling the 10th centile of neonatal weight downward [36]. Nicolaides et al. [36] corrected for this by constructing a compatible Fetal Medicine Foundation BW curve which only included BWs of infants who, as fetuses, had rigidly dated pregnancies in the first trimester and no evidence of any maternal or fetal complications along the way. Since all patients chosen to construct the curve delivered at between 39–40.86 weeks, this excluded EFW data points from fetuses who were not necessarily “normal”. In fact, we found the average BW (2636 g) of those in the SGA group fell between the 5th and 10th percentiles in the Nicolaides curve, rather than the 20th percentile in the Fenton neonatal curve as reported in Table 2.

Another reason for the birthweights to sometimes exceed the 10th percentile neonatally is that we adhered to a type of “intention to treat” concept in fetuses who entered the study with EFWs below the 10th percentile by keeping those in the study who later exceeded this threshold. This allowed our team to follow those who at some time during pregnancy had growth trajectories that rose above the arbitrary 10th percentile threshold and into a subcategory of the lower quartile (10th to 25th percentile). In a large study, Morales-Roselló et al. [37] found that 6.7% of fetuses with BWs between the 10th and 25th percentile had abnormal CPRs (their study definition of “FGR”). This strongly suggests that many fetuses designated as AGA in this category did not live up to their growth potential.

Each strength of the study was balanced with potential liabilities. For example, one strength was that data could be accumulated on 68 patients who had adequate images and Doppler waveforms of all 4 cerebral vessels and 74 patients in whom 3 of the 4 vessels could be evaluated on the last scan before their deliveries. Unfortunately, this left us with different denominators in each vessel category. Additionally, since there was a small number of fetuses with abnormal VA waveforms, the relationships to outcomes in our study will need to be addressed in future studies with, hopefully, larger numbers. Further, although we were able to retrieve outcome data on all infants delivering at the University Hospital, extracting information from records on a few infants delivering at outside hospitals was unsuccessful. This left us with dissimilar denominators in some outcome categories. We adapted our analysis to account for both issues.

Another possible strength was an ability, based on the nature on our referral patient population, to study a cohort heavily weighted towards late-onset FGR, a category less frequently studied. This also was a limitation since a higher risk population which contained more seriously affected FGR fetuses might have yielded even more dramatic results.

A weakness of the study is that one of our goals could only be reached indirectly. The results only inferred a front-to-back process of cerebral dilatation, which appeared to coincide with worsening fetal condition, but this type of progressive response to gradual fetal hypoxia can better be proven via longitudinal studies.

FGR and brain sparing have been linked to hypoxia-related morbidities. Our goal was to search for information provided by lesser-studied cerebral vessels to determine the extent of hypoxia pan-cerebrally. Our results allow us to hypothesize that Doppler of the VA (which may be more specific for a later occurring, more severe phenotype), used in conjunction with the MCA Doppler (a more sensitive, earlier screening modality for brain sparing), may be useful in providing information about the duration and extent of the hypoxia and should provide an incentive for future studies to apply this concept to a larger study group containing both late-onset and early-onset FGR, with emphasis on the identification or prevention of immediate as well as long term morbidities.

## 5. Conclusions

FGR fetuses defined by ISUOG guidelines deliver earlier, have lower birthweights, and have a much higher rate of abnormal cerebral vessel PIs than SGA fetuses. This study suggests that there are varying degrees of brain redistribution, with abnormal posterior brain blood flow being associated with more severe outcomes. We postulate that the VA, used in conjunction with the MCA, may be a useful adjunctive method to identify the duration and severity of FGR.

## Figures and Tables

**Figure 1 jcm-11-04480-f001:**
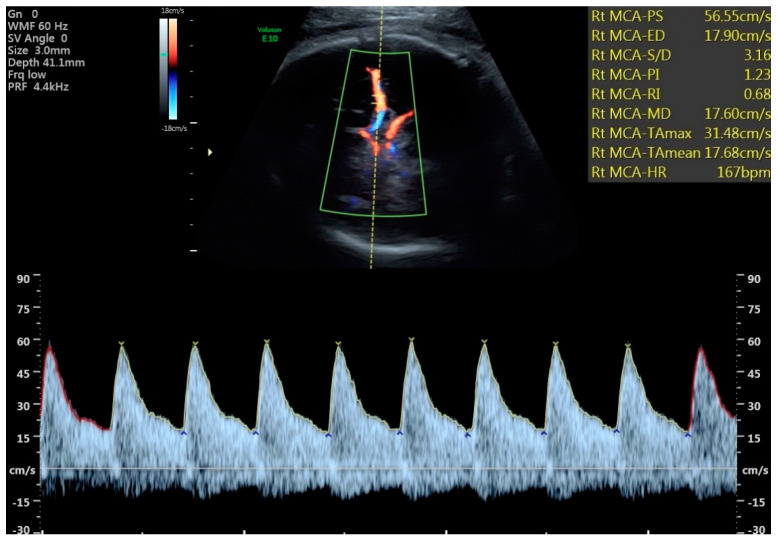
Mid cerebral artery Doppler with brain sparing.

**Figure 2 jcm-11-04480-f002:**
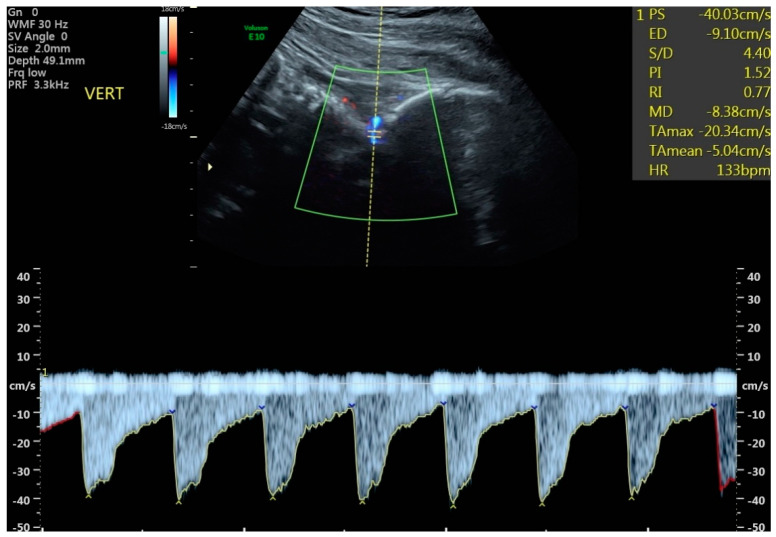
Vertebral artery Doppler. Example of vertebral artery Doppler measurement in a fetus affected by fetal growth restriction.

**Table 1 jcm-11-04480-t001:** Delphi consensus-based definitions for late growth restriction [6,7].

Early FGR: GA < 32 Weeks with No Congenital Abnormalities	Late FGR: GA ≥ 32 Weeks with No Congenital Abnormalities
AC or EFW less than 3rd centile or UA-AEDF	AC or EFW less than 3rd centile
*Or AC or EFW < 10th centile combined with*	*Or at least two of the three:*
Uterine artery PI > 95th centile and/or UAPI > 95th centile	3. AC or EFW less than 10th centile 4. AC or EFW crossing centiles greater than 2 quartiles on growth centiles 5. CPR less than 5th centile *or* UAPI greater than 95th centile *

Abbreviations: GA = gestational age; AC = fetal abdominal circumference; EFW = estimated fetal weight; AEDF = absent end diastolic flow; PI = pulsatility index; UA = umbilical artery; CPR = cerebroplacental ratio. * Growth centiles are non-customized centiles.

**Table 2 jcm-11-04480-t002:** Comparison of patient characteristics in FGR and SGA cohorts.

Cerebral Dopplers between ISUOG Grouping	SGA Mean ± SEM (*n* = 30)	FGR Mean ± SEM (*n* = 51)	*p* Value
Mean MCA PI centile ^†^	41.36% ± 4.86%	41.05% ± 4.46%	0.787
Mean ACA PI centile ^†^	37.80% ± 5.75%	32.38% ± 4.26%	0.377
Mean PCA PI centile ^†^	44.00% ± 4.51%	34.93% ± 4.25%	0.104
Mean UA PI centile ^†^	63.47% ± 3.70%	77.41% ± 3.18%	0.0008 ***
**Clinical Variables**			
GA at Analysis (wks) ^†^	35.82 ± 0.27	35.94 ± 0.22	0.710
GA at Delivery (wks) ^†^	37.84 ± 0.22	37.29 ± 0.23	0.186
Birthweight (g) ^‡^	2636 ± 59.86	2247 ± 65.06	<0.0001 ***
Cesarean Section Rate ^§^	4/26 (15.38%)	10/37 (27.03%)	0.362
Fenton Birthweight (%) ^‡^	20.61% ± 2.88%	6.91% ± 0.78%	<0.0001 ***
NICU Admission ^§^	4/22 (18.18%)	13/36 (36.11%)	0.234

Mann-Whitney U ^†^ or independent samples *t*-tests ^‡^ were used following a normality test. Fisher’s exact tests ^§^ were used for categorical variables. Because standard deviation equations were not published with the VA PI nomogram, we are unable to calculate PI centile for VA Dopplers. Abbreviations: SEM: standard error of the mean; MCA: middle cerebral artery; PI: pulsatility index; ACA: anterior cerebral artery; PCA: posterior cerebral artery; UA: umbilical artery; GA: gestational age; wks: weeks; NICU: neonatal intensive care unit. “* indicates statistical significance (* *p* < 0.05; ** *p* < 0.01; *** *p* < 0.001).”

**Table 3 jcm-11-04480-t003:** Comparison of cerebral Dopplers in FGR and SGA cohorts.

Number of Abnormal Dopplers in ISUOG Groups	SGA (*n* = 30)	FGR (*n* = 51)	*p*-Value
Total # of abnormal Dopplers ^†^ (1 pt each for MCA, ACA, PCA, VA; max: 4 pts/fetus)	4	36	0.055
# of fetuses with ≥1 abnormal cerebral Doppler ^‡^	4/30	15/51	0.113
# of fetuses with MCA PI < 5th centile ^‡^	1/30	10/51	0.047 *
# of fetuses with ACA PI < 5th centile ^‡^	2/27	11/46	0.113
# of fetuses with PCA PI < 5th centile ^‡^	1/29	7/46	0.141
# of fetuses with VA PI < 5th centile ^‡^	0/26	8/43	0.021 *

Comparisons of cerebral dopplers were performed with Mann-Whitney U ^†^ and Fisher’s exact ^‡^ tests. Full Doppler information was not available on all 81 patients; thus, the N varies by Doppler study performed as evident above. Abbreviations: MCA: middle cerebral artery; PI: pulsatility index; ACA: anterior cerebral artery; PCA: posterior cerebral artery; VA: vertebral artery. “* indicates statistical significance (* *p* < 0.05).”

**Table 4 jcm-11-04480-t004:** Comparison of patient characteristics in fetuses with and without abnormal VA Doppler (<5th centile).

Cerebral Dopplers in Fetuses with Normal and Abnormal VA	Pts w/Abnormal VA Dopplers Mean ± SEM (*n* = 8)	Pts w/Normal VA Dopplers Mean ± SEM (*n* = 61)	*p*-Value
Mean MCA PI %ile **^†^**	9.96% ± 5.56%	44.02% ± 3.81%	0.0002 ***
Mean ACA PI %ile **^†^**	3.41% ± 0.56%	37.44% ± 3.62%	<0.0001 ***
Mean PCA PI %ile **^†^**	7.72% ± 3.22%	41.09% ± 3.35%	<0.0001 ***
**Clinical Variables**			
GA at Analysis (wks) **^†^**	34.64 ± 0.71	36.13 ± 0.17	0.023 *
GA at Delivery (wks) **^‡^**	35.22 ± 0.63	37.89 ± 0.14	0.0052 **
Birthweight (g) **^‡^**	1712 ± 151.71	2500 ± 48.85	<0.0001 ***
Cesarean Section Rate ^§^	5/7 (71.43%)	8/49 (16.32%)	0.005 **
Fenton Birthweight (%) **^†^**	4.64% ± 1.37%	13.37% ± 1.80%	0.036 *
Admission to NICU ^§^	5/7 (71.43%)	11/45 (24.44%)	0.023 *

Mann-Whitney U ^†^ or independent samples *t*-tests ^‡^ were used following a normality test. Fisher’s exact tests ^§^ were used for categorical variables. Because standard deviation equations were not published with the VA PI nomogram, we are unable to calculate PI centile for VA Dopplers. For this study, VA PI is assessed as a categorical variable that is either normal (>5th centile) or abnormal (<5th centile). Abbreviations: SEM: standard error of the mean; VA: vertebral artery; MCA: middle cerebral artery; PI: pulsatility index; ACA: anterior cerebral artery; PCA: posterior cerebral artery; GA: gestational age; wks: weeks; NICU: neonatal intensive care unit. “* indicates statistical significance (* *p* < 0.05; ** *p* < 0.01; *** *p* < 0.001).”

**Table 5 jcm-11-04480-t005:** Comparison of patient characteristics in fetuses with and without abnormal CPR (<5th centile).

Cerebral Dopplers in Fetuses with Normal and Abnormal CPR	Pts w/Abnormal CPR Dopplers Mean ± SEM (*n* = 14)	Pts w/Normal CPR Dopplers Mean ± SEM (*n* = 67)	*p*-Value
Mean MCA PI %ile ^†^	7.83% ± 2.14%	48.14% ± 3.42%	<0.0001 ***
Mean ACA PI %ile ^†^	12.78% ± 3.29%	39.06% ± 3.84%	0.0011 **
Mean PCA PI %ile ^†^	21.19% ± 6.16%	42.05% ± 3.44%	0.006 **
# of VA PI < 5th centile ^§^	5/12	3/57	0.003 **
**Clinical Variables**			
GA at Analysis (wks) ^†^	35.68 ± 0.48	35.94 ± 0.18	0.350
GA at Delivery (wks) ^†^	36.73 ± 0.52	37.69 ± 0.16	0.037 *
Birthweight (g) ^‡^	2009 ± 135.31	2492 ± 47.67	0.0001 ***
Cesarean Section Rate ^§^	4/11 (36.36%)	10/52 (19.23%)	0.243
Fenton Birthweight (%) ^†^	3.93% ± 1.05%	14.30% ± 1.71%	<0.0001 ***
Admission to NICU ^§^	5/12 (41.67%)	12/46 (26.09%)	0.307

Mann-Whitney U ^†^ or independent samples *t*-tests ^‡^ were used following a normality test. Fisher’s exact tests ^§^ were used for categorical variables. Because standard deviation equations were not published with the VA PI nomogram, we are unable to calculate PI centile for VA Dopplers. For this study, VA PI is assessed as a categorical variable that is either normal (>5th centile) or abnormal (<5th centile). Abbreviations: SEM: standard error of the mean; CPR: cerebroplacental ratio; MCA: middle cerebral artery; PI: pulsatility index; ACA: anterior cerebral artery; PCA: posterior cerebral artery; VA: vertebral artery; GA: gestational age; wks: weeks; NICU: neonatal intensive care unit. “* indicates statistical significance (* *p* < 0.05; ** *p* < 0.01; *** *p* < 0.001).”

## Data Availability

The data that support the findings of this study are available from the corresponding author upon reasonable request.

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
