# Peer review of "Patterns of Brain Sparing in a Fetal Growth Restriction Cohort"

_jcm, 2022, doi:10.3390/jcm11154480_

Round 1
Reviewer 1 Report
This manuscript reports data on several vessels of the circle of Willis in FGR and SGA fetuses.
The manuscript is interesting and the authors postulate that the VA and the MCA Doppler may be of help in fetuses with an EFW < 10th percentile.
My comments and suggestions to the authors are as follows.
Please add the reference ranges for the Doppler parameters that you used for the different fetal vessels to differentiate between normal vs abnormal. If the Journal has space would be interesting adding a figure with the data plotted on the reference ranges.
Emphasize that your study includes late SGA and FGR. I would clarify this in the Objective paragraph.
Did you get only one measurement per fetus? If the fetuses were studied in more than one occasions what measurement(s) did you report?
Please expand how this manuscript may be helpful in FGR and SGA. Is it important for the physiology/management of these fetuses?
Please clarify the outcome of the fetuses. Admission to the NICU for how many days? Any complications? How many days did the fetuses spend in the Hospital? For how many fetuses were the outcome available? etc. A table would be helpful.
Please clarify
a) the indication(s) of CD
b) the indication for delivery
Perhaps the authors may consider
a) adding the small number of fetuses among the weaknesses.
b) shortening the Comment section and expand how studies like the one here reported may help in the future.
Reviewer 2 Report
The authors presented studies on Doppler blood flow comparison analysis in four fetal intracranial blood vessels from fetal growth-restricted fetuses vs. those small for gestational age. Their results pointed out the importance of Doppler VA evaluation for the identification of fetal growth-restricted fetuses along with the severity of the condition.
The manuscript is clearly written, the results are well presented and the interpretation of data is very appropriate.
Minor points
Abstract: some used acronyms are not written in extension please provide a full explanation (such as MCA and VA).
Results:
It will be interesting to add in table 2, Comparison of Patient Characteristics in FGR and SGA cohorts, the gender ratio and eventually perform the following analysis also considering the sex.
Table 4, “Pts w/ Normal VA Dopplers Mean ± SEM (n=61)” The number of patients reported is the total of the fetus with normal VA, however on this basis, from table 3 the number should be 69, please verify.
Table 4 Clinical Variables, the GA at analysis and delivery is relatively lower than those reported in table 2 which shows general clinical variables of the cohort studied. Please, verify, add details and provide an explanation.
